# MAF: Multi-Aspect Feedback for Improving Reasoning in Large Language Models

**Deepak Nathani , David Wang , Liangming Pan , William Yang Wang**
University of California, Santa Barbara, Santa Barbara, CA
{dnathani, d_wang, liangmingpan, william}@cs.ucsb.edu

## Abstract

Language Models (LMs) have shown impressive performance in various natural language tasks. However, when it comes to natural language reasoning, LMs still face challenges such as hallucination, generating incorrect intermediate reasoning steps, and making mathematical errors. Recent research has focused on enhancing LMs through *self-improvement* using feedback. Nevertheless, existing approaches relying on a single generic feedback source fail to address the diverse error types found in LM-generated reasoning chains. In this work, we propose ***Multi-Aspect Feedback***, an iterative refinement framework that integrates multiple feedback modules, including frozen LMs and external tools, each focusing on a specific error category. Our experimental results demonstrate the efficacy of our approach to addressing several errors in the LM-generated reasoning chain and thus improving the overall performance of an LM in several reasoning tasks. We see a relative improvement of up to 20% in Mathematical Reasoning and up to 18% in Logical Entailment. We release our source code, prompts, and data[1] to accelerate future research.

## 1 Introduction

Recent research in Language Models has focused on augmenting them with external tools (Schick et al., 2023; Paranjape et al., 2023), learning from feedback (Ouyang et al., 2022; Akyurek et al., 2023) and iterative-refinement (Madaan et al., 2023; Paul et al., 2023; Shinn et al., 2023). Iterative-Refinement has been a necessary tool in human evolution and problem-solving. Moreover, humans seek feedback from *domain-specific* knowledge sources. For instance, an architect tasked with creating an environmentally friendly and structurally sound building design will require targeted feedback from civil engineers for structural integrity, and sustainability experts for eco-friendly

design principles. However, previous works on iterative refinement overlook this requirement and collect *generic* feedback from multiple sources or collect a single one from the Language Model itself.

In this work, we first investigate whether the use of generic feedback is a bottleneck in addressing the diverse range of errors present in the reasoning chains generated by LMs. We posit that utilizing generic feedback for a broad spectrum of errors may result in vague or non-actionable feedback for specific errors. This is due to two main factors: (i) the inclusion of multiple error categories within a single prompt, which not only increases the size of the prompt but also poses a challenge to current models that have a limited context length and struggle with long texts, and (ii) the model is burdened with identifying a multitude of error types in a single instance, which degrades the quality of the generated feedback.

To surmount these challenges, we introduce Multi-Aspect Feedback (MAF), a novel general-purpose iterative refinement framework that employs a collection of specialized feedback modules, including pre-trained LMs and external tools, each tailored to address a specific error category. Our framework includes a base model to generate an initial solution, multiple feedback modules, and a refiner model to revise the solution using feedback. Contrary to previous works, our feedback modules can use one of two refinement strategies: `Eager Refinement` and `Lazy Refinement` §3.4. When using the Eager-Refine mode, the solution is revised immediately before moving on to the next feedback, however, for Lazy Refinement, we first collect feedback from multiple feedback sources and then refine the solution using this collective feedback.

In devising MAF, we first classified errors in LM-generated reasoning chains based on Golovneva et al. (2022) and also identified some new error

---

[1] Our source code can be found here

Figure 1: Overview of the Multi-Aspect Feedback framework. We loop over steps 3 and 4 until we reach a final solution $O'$ or up to a set number of iterations.

categories. Subsequently, we decoupled our feedback modules such that each module focuses on a single error category. This strategy not only elevates performance but also allows one to leverage specialized tools tailored for distinct error types, such as utilizing a code interpreter for syntax errors in generated Python programs instead of relying on generic LM feedback. Through a modular design, we ensure that each module of our framework is replaceable depending on the task. We conduct extensive experimentation on a variety of complex reasoning tasks such as Mathematical Reasoning (Shi et al., 2023), Logical Reasoning (Dalvi et al., 2021), and Multi-Hop Question Answering (Dua et al., 2019), and establish that MAF effectively addresses the challenges previously discussed, and significantly enhances the performance of Language Models in complex reasoning tasks. We compare our method against the Base LMs and Self-Refine (Madaan et al., 2023), which also implements iterative refinement. During our experiments, we find that `Over-Refining` §4 the solution can lead to worse performance, and thus we treat the number of iterations as a hyperparameter. We also propose an Oracle Verifier setting §4, where we assume we have access to an "answer verifier" and stop the refining process once we have reached the correct answer. To avoid unfair comparisons,

we test Self-Refine under this setting and find that MAF provides actionable and specific feedback.

To summarize, the main contributions of this work are as follows:

1. We propose a novel framework that decouples the feedback generation process for different error categories. This allows us to use error-specific tools and LMs. Moreover, the proposed framework is modular and all the modules are Plug-and-Play. This allows for better flexibility on new tasks and the usage of advanced models as they become available.

2. Additionally, we propose two different types of refinement strategies, `Eager Refinement` and `Lazy Refinement`. Eager Refinement allows immediate revision of a solution by a feedback module to avoid conflicts, while Lazy Refinement improves efficiency by performing revisions from multiple feedback modules together.

3. We show that our framework outperforms the base Language Models and similar Iterative Refinement baselines on several complex reasoning tasks such as Math reasoning (GSM-IC), Logical Reasoning (Entailment Bank), and Multi-hop Question Answering (DROP).

## 2 Related Work

**Chain-of-Thought Reasoning** There has been a plethora of research on prompting Large Language Models to improve their reasoning capabilities. Wei et al. (2023) found that prompting the models to generate a reasoning chain in few-shot setting before solving a problem improves the performance by a huge margin. Further, Kojima et al. (2022) found that in zero-shot setting prefixing the solution generation with *Let's think step-by-step* has the same effect as generating intermediate reasoning chain and improves the performance. Following this work, Wang et al. (2022) proposed sampling multiple outputs from the model and selecting the final answer by majority voting. Zhou et al. (2023) also showed that decomposing the original question into smaller sub-problems can give a performance boost. Madaan et al. (2022) and Chen et al. (2022) showed that models trained on code can be used to generate a *program-of-thought* similar to a *chain-of-thought*, which enables the model to use a language interpreter for translating mathematical calculations into code. In this work, we use LMs' in-context learning abilities to implement the LM-based feedback and refiner modules.

**Tool Augmented LLMs** However, even with these advanced prompting techniques, the LMs still fail for problems that require external knowledge and are still prone to problems such as hallucination and incorrect reasoning. To circumvent these issues, several recent works have introduced the use of external tools such as calculator, code interpreter, knowledge retrieval (Karpas et al., 2022; Chen et al., 2022; Schick et al., 2023). Yang et al. (2023); Patil et al. (2023); Shen et al. (2023) show that one can teach LLMs to use tools by generating API calls. Li et al. (2023) also released a large dataset to enable research in the field of augmented Language Models. Hao et al. (2023) propose to represent each tool as a token and learn tool-specific embeddings to increase the robustness of these language models for using external APIs. However, reasoning is an iterative task and it is difficult to find a plausible answer for several problems in one-shot even when these LLMs are augmented with tools. This has inspired some recent works to use iterative refinement frameworks. In our work, we use external tools for generating feedback for certain error categories such as Programming Syntax Errors, Calculator for mathematical equations etc.

| Task | Feedback Type | Type | ER |
|------|---------------|------|----|
| Math Reasoning | Programming Syntax | Interpreter | ✓ |
| | Variable Naming | OpenAI | ✓ |
| | Redundancy | OpenAI | ✗ |
| | Commonsense | OpenAI | ✗ |
| | Missing Step | OpenAI | ✗ |
| Logical Reasoning | Redundancy | OpenAI | ✗ |
| | Repetition | OpenAI | ✗ |
| | Hallucination | OpenAI | ✗ |
| Question Answering | Redundancy | OpenAI | ✗ |
| | Factuality | OpenAI | ✗ |
| | Commonsense | OpenAI | ✗ |
| | Missing Step | OpenAI | ✗ |

Table 1: Feedback modules used for each task and their types. OpenAI type modules use a LLM to provide feedback. Feedback Modules with `Eager-Refine (ER)` enabled, refine the solution without waiting for feedback from the other feedback modules. (§3.4).

**Learning from Feedback** Schick et al. (2022) took the first step towards iteratively fixing the problems and introduced the idea of training multiple instances of the same model to assist in different stages of problem-solving. It was introduced as a collaborative editing framework. (Madaan et al., 2023) proposed to use the same model for generating initial solution, feedback generation, and refiner. Furthermore, recently there have been multiple works exploring the use of natural language feedback to improve performance. Shinn et al. (2023) proposes converting different types of feedback into natural language and storing the feedback in a memory buffer to revisit later. Akyurek et al. (2023) train a critique generator to maximize the end-task performance of a larger base model using reinforcement learning. Paul et al. (2023) proposes a framework to fine-tune LMs for generating intermediate reasoning steps while interacting with a trained critique model to bridge the gap between small and large models. Recent research (Wu et al., 2023) has also demonstrated the benefits of incorporating decoupled fine-grained feedback for each error category by using a set of fine-tuned reward models. We defer to Pan et al. (2023) for a comprehensive discussion of research on automated feedback generation and correction.

Despite these advancements, there is a lack of empirical evidence supporting the effectiveness of decoupled multi-aspect feedback for *iterative refinement*. Our work addresses this gap by demonstrating that a generic feedback module, is insuffi-

cient to address the diverse range of potential errors in language model responses. To overcome this limitation, we propose a suite of feedback modules, each specifically targeting a particular error category and providing detailed feedback to improve both reasoning and solution quality.

## 3 MAF: Multi-Aspect Feedback

### 3.1 Overview

In this work, we present an iterative refinement framework with an explicit focus on decoupling feedback generation for different error categories. This approach allows us to systematically address different error types in the generated solutions. Our proposed framework has three crucial components: a base language model $M$ that generates an initial solution $O$. A collection of $n$ feedback modules $\{f_0, f_1, f_2...f_n\}$, each focusing on a single error category, collectively these modules generate a multi-aspect feedback $F$. And a Refiner $R$ that generates refined solution $O'$ based on initial solution $O$ and feedback $F$.

While there are other works that also use iterative refinement (Madaan et al., 2023; Akyurek et al., 2023; Peng et al., 2023), to the best of our knowledge, we are the first to explore the effect of decoupling different types of feedbacks. Taking inspiration from ROSCOE (Golovneva et al., 2022), we categorize feedback into ten distinct categories: arithmetic, programming syntax, variable naming, missing step, coherency, redundancy, repetition, hallucination, commonsense, and factuality. Definitions of these error categories are provided in Appendix A. Moreover, a feedback module can be a tool such as a code interpreter for `syntax feedback`, a calculator for `arithmetic feedback`, a knowledge graph for `factuality or commonsense feedback`, a Language Model, or even a fine-tuned model. It is important to note that the feedback generation process is not limited to these categories and can be extended to include other categories as well. The overall process is illustrated in Figure 1 and Algorithm 1.

### 3.2 Initial Generation

We use a large model such as GPT3.5[2], GPT4 (OpenAI, 2023), to generate an initial solution. However, generating just one solution isn't ideal for the reasoning process since reasoning is an iterative

[2]https://openai.com/blog/chatgpt

process. The solution often needs to be refined over time with every iteration bringing us closer to the correct answer. We follow the same principle in this work, where we initially produce a solution and then proceed to refine it based on actionable feedback.

### 3.3 Feedback Modules

Feedback generation is an involved task, providing a comprehensive list of common issues encountered in model outputs. The feedback generation is accomplished through a variety of tools known as feedback modules, which may include external tools, frozen LLMs, fine-tuned models, and scorers. These modules are used to provide actionable feedback based on the initial solution.

Each type of feedback module is suited to address specific types of errors. For instance, an external tool like a code interpreter would be ideal for providing feedback on syntax errors in code, while a fine-tuned model could provide more nuanced feedback on issues such as redundancy or hallucination. This decoupled approach allows us to address errors in a more targeted manner, improving the overall quality of the refined solution. Table 1 shows all the feedback modules used in our work.

Furthermore, it's worth noting that merely specifying the error categories in a single prompt doesn't yield satisfactory results. We see two main reasons for this, firstly because the model is tasked with focusing on multiple errors simultaneously and secondly, as the number of error categories increase, the context length increases as well which results in high-quality feedback for the first few error categories and a steep decline in quality for the rest. While some of the previous works (Wu et al., 2023; Paul et al., 2023) have explored using fine-grained feedback and achieved promising results.

Moreover, our feedback modules can choose between two refinement strategies: `Eager-Refine` or `Lazy-Refine`. We discuss this distinction in more detail in the next section.

### 3.4 Refinement

In this work, we reuse the initial model $M$ as a Refiner. During the refining phase, the refiner is given a solution and multi-aspect feedback and asked to revise the solution according to feedback. We found that these large models are proficient at the refining task, demonstrating a marked improvement in the quality of the final solution compared

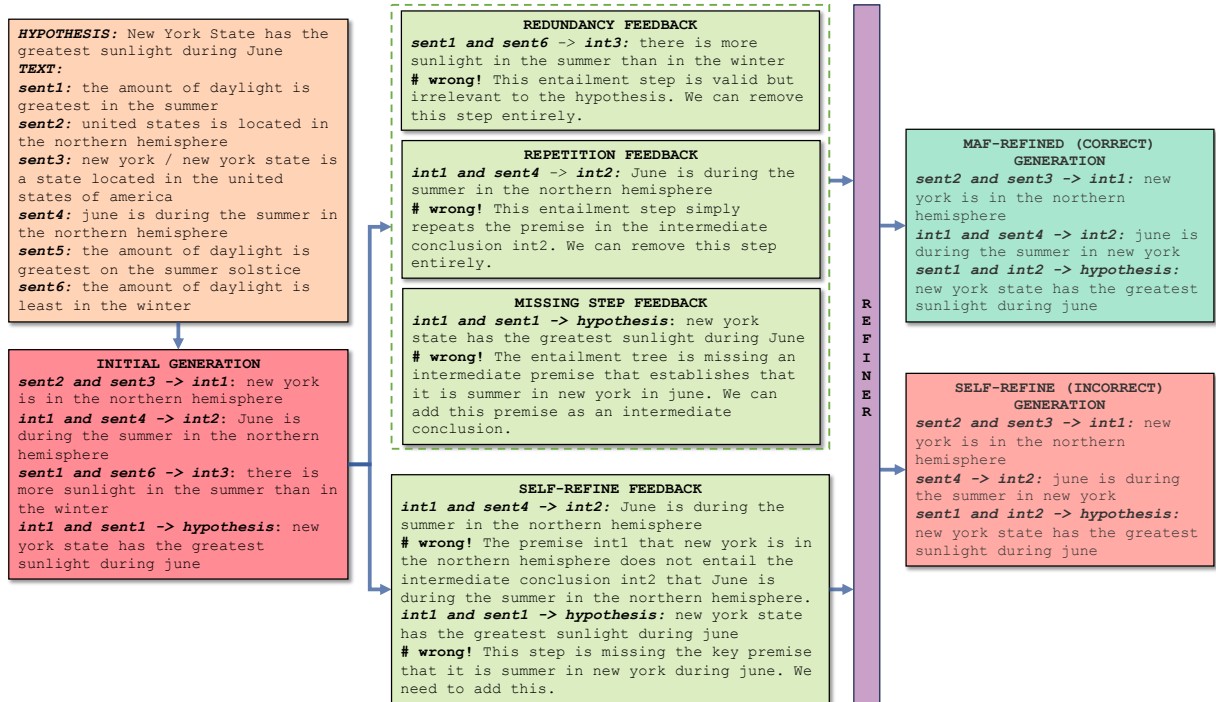

Figure 2: Comparison of the feedback generated by MAF and Self-Refine.

to the initial output.

As mentioned in the previous section, we use two refinement strategies: *Eager-Refinement* and *Lazy-Refinement*. The eager-Refinement approach is used for feedback types that can cause conflicts during refinement. An example of such as feedback module would be *Variable Naming* (VN), which is used to correct variable names in the generated code. This module can cause conflicts with others because when the other modules are referencing a variable that is supposed to be changed according to the VN feedback and can render the program inexecutable if refined incorrectly. Whereas, in the lazy refinement strategy feedback from the multiple modules is concatenated together along with the appropriate error categories to make a single multi-aspect feedback. This collective feedback is then passed to the Refiner model in order to get a revised solution. Another advantage of this approach is the increased efficiency by refining once for multiple errors and the added flexibility.

However, we also found that the smaller open-source models like LLaMA (Touvron et al., 2023), Alpaca (Taori et al., 2023) and Vicuna[3] often fail to adhere to the feedback provided thus making the refinement process ineffective. These smaller models also have smaller context lengths which

make it difficult to include all the feedback in the prompt. To address this issue, we use a Selective Summarization approach. We only select parts of feedback that point to a problem. Figure 2 shows the "summarized" feedback generated by our approach. This simple selective summarization approach makes the feedback succinct and also proves to be less distractive to the models during the refining phase. This approach allows us to effectively combine all feedback together and use models with smaller context lengths to some extent.

## 4 Experiments

### 4.1 Datasets and Metrics

**Mathematical Reasoning** The GSM8K dataset, presented by Cobbe et al. (2021), is a comprehensive compilation of high-quality grade school math problems. GSM8K has proven to be a great resource for testing LLMs on mathematical word problems, however, the performance on this dataset has been saturated for a while.

To avoid that problem, we conduct experiments on a harder variant of this dataset, GSM-IC (Grade-School Math with Irrelevant Context), that was introduced by Shi et al. (2023). We run our main experiments on a randomly sampled subset of 500 problems from GSM-IC and use % solve rate as

---

[3]https://lmsys.org/blog/2023-03-30-vicuna/

**Algorithm 1** MAF algorithm

---

**Require:** Input $x$, model $M$, Refiner $R$, number of iterations $T$
**Require:** $n$ Eager-refine $\{s_1, s_2, ...s_n\}$ and $m$ Lazy-Refine $\{p_1, p_2, ...p_m\}$ Feedback Modules

1: Initialize output $y_0$ from $M$
2: **while** $i < T$ **do**
3:     **for** $j \leftarrow 1$ to $n$ **do**            ▷ Eager-refine Feedbacks
4:        $fs_j \leftarrow generate\_feedback(s_j, y_i)$       ▷ Generate Feedback
5:        **if** $fs_j$ indicates revision is required **then**
6:           $y_i \leftarrow revise(R, fs_j, y_i)$           ▷ Revise solution
7:     $F \leftarrow$ ””            ▷ Initialize empty feedback
8:     **for** $k \leftarrow 1$ to $m$ **do**           ▷ Lazy-refine Feedbacks
9:        $fp_k \leftarrow generate\_feedback(p_k, y_i)$
10:        **if** $fp_j$ indicates revision is required **then**
11:           $F \leftarrow F + fp_k$
12:     **if** $F$ not empty **then**
13:        $y_{i+1} \leftarrow revise(R, fp_k, y_i)$
14:     **else**
15:        $y_{i+1} = y_i$

---

the metric.

**Logical Reasoning** EntailmentBank, as described by Dalvi et al. (2021), is a dataset that contains multistep entailment trees. We use the validation set of Task 1 provided by the authors for our experimentation. For the metrics, we do not use the automated metrics provided by the original work because these metrics expect the trees to be similar in structure to gold trees. However, that is not a fair comparison because we find that there exist multiple correct entailment trees for a given hypothesis and information. Thus, we conduct a human evaluation and ask humans to evaluate if the hypothesis can be entailed from the predicted tree.

**Question Answering** DROP (Dua et al., 2019) is a question-answering dataset. The Discrete Reasoning Over the Content of Paragraphs (DROP) dataset is designed for complex question-answering tasks that require multi-step reasoning over text passages. It presents a valuable benchmark for our iterative refinement framework, as the multi-step nature of the questions offers ample opportunities for generating feedback and refining to guide the model toward a correct solution. We use output parsing for answers and use % correct answers as the final metric.

### 4.2 Baselines

In this work, we focus on comparing our method with the Base LMs using the Ope-

nAI API and a recently proposed iterative refinement framework, Self-Refine (Madaan et al., 2023). To provide a fair comparison and avoid randomness in the generated answer, we used Greedy Decoding for all our experiments.

We follow (Madaan et al., 2023) and use 8-shot Program of Thought (Chen et al., 2022) for GSM-IC since the Python program written by the model acts as a "calculator" for mathematical equations.

The input for Entailment Bank includes a hypothesis and the supporting text, and we are limited by the context length of the current models. Hence, we use 4-shot prompting for this dataset. We use standard few-shot prompting (Brown et al., 2020), because the Entailment Tree itself acts like a reasoning chain.

Similarly, the examples in the DROP dataset have a passage and an accompanying question, so we use a 3-shot Chain of Thought Wei et al. (2023) prompting. We also provide an instruction specifying that the model should select either a number, date, or span from the passage to answer the question as shown in Appendix C.

For Self-Refine, we use the prompts provided by the authors for GSM8K in their work for GSM-IC and write our own prompts for DROP and Entailment Bank. Self-Refine has three modules, initial generation, feedback, and refiner. We use the same parameters and prompting strategy as the corresponding baseline for the initial generation. The feedback module and refiner module are imple-

| Model | EB | GSMIC | GSM8K | DROP |
|---|---|---|---|---|
| GPT3.5 | **56.1** | 76.2 | 69.2 | **72.3** |
| +SR | 53.5 ↓2.6 | 77.0 ↑0.8 | 69.2 ↓0.0 | 62.0↓8.3 |
| +SR★ | 54.5 ↓1.6 | 87.0 ↑10.8 | 77.4 ↑8.2 | 77.5 ↑4.2 |
| +MAF | 54.5 ↓1.6 | **77.4** ↑1.2 | 69.8 ↑0.6 | 66.2 ↓6.1 |
| +MAF★ | 60.4 ↑4.3 | 91.4 ↑15.2 | 73.4 ↑4.2 | 76.4 ↑4.1 |
| ChatGPT | 60.4 | 72.0 | 71.8 | **70.7** |
| +SR | 65.8 ↑5.4 | 76.0↑4.0 | **74.6** ↑2.8 | 45.5↓25.2 |
| +SR★ | 67.4↑7.0 | 78.0↑6.0 | 79.4 ↑7.6 | 73.2↑2.5 |
| +MAF | **68.4**↑8.0 | **77.8**↑5.8 | 73.2 ↑1.4 | 67.9↓2.8 |
| +MAF★ | 71.7 ↑11.3 | 82.8↑10.8 | 76.6 ↑4.8 | 72.7↑2.0 |

Table 2: Experimental results for Entailment Bank (EB), GSMIC, GSM8K, DROP dataset as described in §4. *SR* represents Self-Refine (Madaan et al., 2023), and *MAF* represents our method. ★ represents the Oracle Verifier setting (§4.4). The best score for standard setting is in **bold** and underlined for the Oracle Verifier setting.

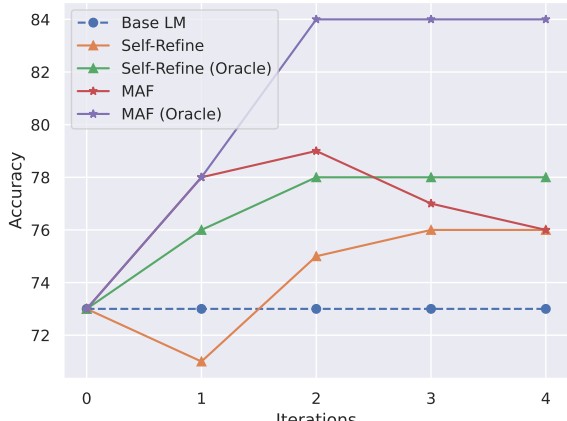

Figure 3: Accuracy for baselines and MAF on GSM-IC using ChatGPT under Standard and Oracle Verifier setting §4.4.

mented using the standard few-shot prompting with 3 in-context examples for all datasets. Further details about the implementation of these baselines can be found in Appendix C.

### 4.3 Implementation

We implement MAF (Multi-Aspect Feedback) following the basic structure defined in §3 and Algorithm 1. The iterative refinement process continues until we reach the desired output quality or a task-specific stop criterion, up to a maximum of 4 iterations. Our method includes an initial generation, refiner, and feedback modules as shown in Table 1. The initial generation module uses the same parameters and prompting strategies as the corresponding baseline. Our Eager-Refine module uses 3-shot standard prompting and our Lazy-Refiner, which includes feedback from multiple modules, uses a 2-shot standard prompting approach. Our OpenAI-based feedback modules also use 3-shot standard prompting following (Madaan et al., 2023) with an error-specific instruction.

To provide a fair comparison, we use the same prompts for Baselines and Self-Refine wherever possible. We use Greedy Decoding for all our OpenAI-based[4] modules to avoid any randomness. As mentioned in §3.3, we use a *selective summarization* approach to fit all the lazy-refine feedbacks in our refiner module. We use a basic rule-based

strategy to summarize feedback. Since our feedback modules are instructed to inspect each intermediate step, they also include the steps with no mistakes in the generated feedback. However, feedback on these lines is not useful as such because there is no change in those steps. Thus we look for the steps with feedback "looks good" and remove those steps. This simple approach helps us increase the efficiency of our method by being able to include multiple feedbacks in our lazy-refiner.

### 4.4 Results

In this section, we discuss the performance of our method and compare it to the baseline. We also discuss an ablation experiment studying the contribution of each feedback module. The main results of this work are shown in Table 2. We find that our method can outperforms the base LMs and Self-Refine on a diverse set of reasoning datasets. For MAF, we report the results after 2 iterations, and following Madaan et al. (2023) report the Self-Refine results after 4 iterations.

**GPT3.5 vs ChatGPT** We found that GPT3.5 can generate better feedback if an error is present in the solution, however often points out spurious errors even when they are not present. On the other hand, ChatGPT is more conservative in its feedback generation and often fails to detect the error even in an erroneous solution. Because of this conflicting behavior from the two models, our method is able to achieve similar performance using both models,

---

[4]Our experiments with GPT3.5 (Text-Davinci-003) and ChatGPT (gpt-3.5-turbo) are conducted using OpenAI API between April-2023 and August-2023.

even though ChatGPT is generally considered to be a better model than GPT3.5.

**Over-Refining** Due to the behavior described above, we face the problem of `Over-Refining`. This means that once we have reached an optimal solution for a reasoning problem, forcing the LM to refine it further deteriorates the performance and the quality of the reasoning chain. As shown in Table 2 and Figure 3, Self-Refine (Madaan et al., 2023) also suffers from this problem in the DROP dataset. Moreover, all the other iterative refinement framework such as Self-Refine uses a stop condition and stop the refining process if the generated feedback does not point out any problem. We however cannot take advantage of this stop condition because of the interplay between multiple feedback modules. For example, let's say our Missing Step module stops at iteration $k$ due to no missing steps at that point in time since the other feedback modules continue to refine the solution and self-refinement is an imperfect process, it might introduce a Missing Step error in further iterations.

Hence, we treat the number of iterations as a hyperparameter and find that 2 iterations work best for our method.

**Oracle Verifier** Since the main focus of this work is to improve the incorrect initial generations from the model, we also evaluate the models under an Oracle Verifier setting. In this setting, we assume access to an "Oracle Verifier" which can judge the final answer generated by the model. If the final answer is correct, we stop the refinement process for that test sample, otherwise, we let the model continue refining the solution. It is important to note however that the model is not privy to this verification, and hence can still stop further refinement by not generating actionable feedback in the next iteration. Under this setting, we report results for both Self-Refine and MAF after 4 iterations. Even under this modified setting, we see that our method can outperform Self-Refine because we generate diverse feedback and thus can improve more solutions in the GSM-IC dataset.

## 4.5 Ablation

In this section, we perform two ablation studies. Firstly, we analyze the impact of each feedback module, identifying those with the most significant effect on performance. Secondly, we compare the outcomes of our two proposed refinement strategies: `Lazy` vs `Eager` Refinement. The ablation studies are conducted on the GSM-IC dataset. Due to resource constraints, we conduct the following ablation studies on a smaller random subset of size 100.

| Model | Standard | Oracle |
|---|---|---|
| ChatGPT | 73.0 | — |
| +MAF | 79.0 | 84.0 |
| −Variable Naming (VN) | 79.0 ↓0.0 | 83.0 ↓1.0 |
| −Redundancy (Red) | 70.0 ↓9.0 | 80.0 ↓4.0 |
| −Commonsense (Com) | 73.0 ↓6.0 | 80.0 ↓4.0 |
| −Missing Step (MS) | 79.0 ↓0.0 | 84.0 ↓0.0 |
| MAF (VN, Red, Com) | 79.0 | 85.0 |

Table 3: This table shows the contribution of each feedback module in MAF for the GSM-IC dataset. − symbol in front of a module denotes the accuracy of our method after removing that feedback module. MAF (VN, Red, Com) shows the accuracy of our method using only the best-performing feedback modules.

**Contribution of different Feedback Modules** We test the contribution of each feedback module by removing that module and calculating the accuracy under standard and Oracle verifier settings. While the standard setting highlights the potential negative impacts of some of the modules, the oracle verifier setting highlights the *absolute* contribution of each module. This highlights an important finding that using the error categories is paramount to gaining performance. Results for this ablation study can be found in Table 3.

We also calculate the accuracy of our method when using the three best-performing modules as the only source of feedback. We found that this does recover the performance of our method. Even though Variable Naming and Missing Step modules do not affect MAFs performance by a huge margin, it still makes our method more robust to a possible distribution shift. Moreover, the Variable Naming module's main contribution is not increasing the performance, but rather to increase the readability of the code and not confuse users with unclear names.

Note that we did not remove the Programming Syntax checker module as we use Program of Thoughts which requires a Python interpreter.

| Model | Standard | Oracle |
|---|---|---|
| GPT3.5 | **71.0** | 71.0 |
| +MAF | 67.0 ↓4.0 | **85.0** ↑14.0 |
| +Only Lazy-Refine | 66.0 ↓5.0 | 81.0 ↑10.0 |
| +Only Eager-Refine | 66.0 ↓5.0 | **85.0** ↑14.0 |
| ChatGPT | 73.0 | 73.0 |
| +MAF | **78.0** ↑5.0 | **81.0** ↑8.0 |
| +Only Lazy-Refine | 69.0 ↓4.0 | 80.0 ↑7.0 |
| +Only Eager-Refine | 73.0 ↓0.0 | 80.0 ↑7.0 |

Table 4: Accuracy on GSM-IC when using different refinement strategies under Standard and Oracle settings. *MAF* shows the performance of our method when using both Lazy and Eager refine in tandom. *Only Lazy-Refine* means all feedback modules use Lazy-Refine and similarly *Only Eager-Refine* means all modules use Eager-Refine strategy. The best score for each setting is in **bold**.

**Lazy vs Eager Refinement** To illustrate the complementary nature of our two proposed refinement strategies, we evaluate the performance of our iterative refinement framework using all feedback modules in either Lazy or Eager mode. The results are presented in Table 4, which showcases the performance of our framework under different refinement settings. In this table, 'MAF' corresponds to the results obtained by combining both Lazy and Eager Refinement strategies, as defined in Table 1, while 'Only Lazy/Eager-Refine' displays the results of our framework when utilizing only one type of refinement strategy.

The results clearly demonstrate that our framework, which leverages a combination of eager and lazy refinement, consistently matches or outperforms using either strategy in isolation, across both the standard and oracle verifier settings.

Practical considerations also favor the use of a hybrid approach incorporating both eager and lazy feedback. Relying solely on lazy feedback can lead to a situation where multiple feedback categories are condensed into a single prompt. Despite our feedback summarization technique, this can result in the iteration prompt exceeding the context window limit of many widely available models. Conversely, exclusively employing eager feedback may result in rewriting the solution for each feedback module, leading to high token usage. While using all modules in Eager-Refine mode can closely approach the performance of 'MAF' (as shown in Table 4), it is not scalable when dealing with a large number of feedback modules.

## 5 Conclusion

In this work, we present Multi-Aspect Feedback (MAF), a novel iterative refinement framework that decouples the feedback modules and takes advantage of error-specific tools to generate feedback. We demonstrate the performance of our framework on a set of diverse reasoning tasks and compare it with other iterative refinement baselines.

Contrary to previous works, we found that *Over-Refinement* can be a problem in iterative refinement frameworks since models are not certain if their own answer is correct. Our work also draws focus on the necessity to devise better feedback methods and call for augmenting Language Models with them. We hope this work will inspire further research in this area and to this end, we make all our code, data, and prompts available.

## Limitations

The main limitation of our approach is that the base models need to have sufficient in-context learning abilities to process the feedback and refine the solution. Even with in-context learning abilities, these models are not perfect and thus can still make mistakes while refining the solution even when correct feedback is given.

All the experiments conducted in this work use large powerful models provided by OpenAI. We find that open-source LMs such as Vicuna, and Alpaca can generate decent initial solutions but are not capable of refining their own solutions. Thus, we leave the investigation of improving open-source models to future work.

Another limitation inherent in our approach is the reliance on a fixed set of Feedback Modules. Our method necessitates the pre-selection of feedback modules before execution, which in turn demands human intervention to determine the appropriate feedback categories for each new dataset or domain. Future research could explore novel methods that can dynamically and autonomously determine the most suitable feedback modules for specific problems in real-time.

## Ethics Statement

The experiments in this work were performed with models that are not open-sourced and are ever-changing. Moreover, these models are expensive to use, and thus research using these models requires

an enormous amount of funding. The existing literature lacks details about the datasets that are being used to train these huge models or the filtering mechanism that is being used to clean the polluted corpora.

Furthermore, there is always a possibility for bad actors to use our method to generate more toxic or harmful text. Our approach does not guard against this.

## Acknowledgements

This work was supported by the National Science Foundation award #2048122. The views expressed are those of the authors and do not reflect the official policy or position of the US government. Additionally, we thank our reviewers for their detailed and useful feedback.

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

## A   Error Categories

In this section we define all the error categories that are implemented in this work. Table 5 shows the error categories introduced by Golovneva et al. (2022).

### A.1   Programming Syntax Feedback

Programming syntax feedback module is implemented as Python Interpreter. This module aims to fix any syntax errors in the generated code. This particular module benefits from a Eager-Refinement approach.

### A.2   Variable Naming Feedback

Good variable names in a code can improve the readability of the code and potentially can improve model's own understanding of the program in further iterations. Variable Naming Feedback is another module which benefits from an eager-refine approach.

### A.3   Redundancy Feedback

Redundant information is any information included in the reasoning that doesn't help answer the question. This additional information may distract the model from correctly answering and should thus be removed.

### A.4   Commonsense Feedback

Commonsense reasoning errors are errors about any relation or knowledge that is should be known from general world such as "all ducks are birds".

### A.5   Missing Steps

Missing steps errors are any gaps in reasoning or missing information that prevent the reasoning chain from being correct. This also identifies the model saying that the question is unanswerable as a missing step error because that means additional reasoning steps are needed to answer the question from the passage.

### A.6   Factuality Feedback

Factuality errors are cases where the answer reasoning states infactual information. This could be information that contradicts information given in the passage or hallucinated.

### A.7   Hallucination Feedback

LLMs are prone to hallucination, however, it has been shown that sampling from a LLM multiple times and then selecting the majority answer can improve the results. Thus Hallucination feedback aims to fix any hallucinated facts in the initial generation. This module can be improved by using an external tool such as a Knowledge Source instead of a LLM.

## B   Implementation Parameters

Table 6 provides parameters used for Self-Refine and Table 7 provides parameters used for MAF.

## C   Few-Shot Prompt Examples

We add samples for all the prompts used in our work. Complete prompts can be found in our source code. Note that for all feedback prompt examples, there is at least one example with no errors, in which case the feedback will state that there are no errors and the reasoning is correct. This helps decrease the likelihood that the model identifies an error in a solution that is actually correct.

| Error Type | Definition |
|---|---|
| **Programming Syntax** | Syntax errors in code |
| **Arithmetic** | Error in math calculations |
| **Grammar** | Faulty, unconventional, or controversial grammar usage |
| **Coherency** | Steps contradict each other or do not follow a cohesive story |
| **Variable Naming** | Variable names in a program don't give full information or are wrong |
| **Repetition** | Step paraphrases information already mentioned in previous reasoning steps |
| **Hallucination** | Information is not provided in the problem statement and is irrelevant or wrong |
| **Commonsense** | Model lacks relations that should be known from the general world (e.g., "1 dozen = 12") |
| **Factuality** | Information about an object (i.e. quantity, characteristics) or a named entity doesn't match with the input context |
| **Missing Step** | The content of the generated reasoning is incomplete and lacks the required information to produce the correct answer |
| **Redundancy** | Explanation contains redundant information, which even though might be factual, is not required to answer the question |

Table 5: Error types as defined in Golovneva et al. (2022). Each error category is defined for a single step in the reasoning chain.

| Dataset | Base LM | Feedback | Refiner |
|---|---|---|---|
| EB | 300 | 600 | 600 |
| GSM-IC | 300 | 600 | 600 |
| GSM8K | 300 | 600 | 600 |
| DROP | 450 | 600 | 800 |

Table 6: Value of Maximum Number of Tokens parameter for Self-Refine on various datasets.

| Dataset | Base LM | Feedback | Refiner |
|---|---|---|---|
| EB | 300 | 600 | 600 |
| GSM-IC | 300 | 600 | 600 |
| GSM8K | 300 | 600 | 600 |
| DROP | 450 | 600 | 800 |

Table 7: Value of Maximum Number of Tokens parameter for MAF on various datasets.

```
# Q: Jason had 20 lollipops. He gave Denny some lollipops. Now Jason has 12 lollipops. How many lollipops did
Jason give to Denny?
# solution using Python:

def solution():
    """Jason had 20 lollipops. He gave Denny some lollipops. Now Jason has 12 lollipops. How many lollipops
did Jason give to Denny?"""
    jason_lollipops_initial = 20
    jason_lollipops_after = 12
    denny_lollipops = jason_lollipops_initial - jason_lollipops_after
    result = denny_lollipops
    return result

# Q: There are 15 trees in the grove. Grove workers will plant trees in the grove today. After they are done,
there will be 21 trees. How many trees did the grove workers plant today?
# solution using Python:

def solution():
    """There are 15 trees in the grove. Grove workers will plant trees in the grove today. After they are
done, there will be 21 trees. How many trees did the grove workers plant today?"""
    trees_initial = 15
    trees_after = 21
    trees_added = trees_after - trees_initial
    result = trees_added
    return result

# Q: Shawn has five toys. For Christmas, he got two toys each from his mom and dad. How many toys does he have
now?
# solution using Python:

def solution():
    """Shawn has five toys. For Christmas, he got two toys each from his mom and dad. How many toys does he
have now?"""
    toys_initial = 5
    mom_toys = 2
    dad_toys = 2
    total_received = mom_toys + dad_toys
    total_toys = toys_initial + total_received
    result = total_toys
    return result

# Q: There were nine computers in the server room. Five more computers were installed each day, from monday to
thursday. How many computers are now in the server room?
# solution using Python:

def solution():
    """There were nine computers in the server room. Five more computers were installed each day, from monday
to thursday. How many computers are now in the server room?"""
    computers_initial = 9
    computers_per_day = 5
    num_days = 4  # 4 days between monday and thursday
    computers_added = computers_per_day * num_days
    computers_total = computers_initial + computers_added
    result = computers_total
    return result

...
```

Figure 4: Initialization prompt for Mathematical Reasoning

```
def solution():
    """Kelly is grocery shopping at a supermarket and is making sure she has enough in her budget for the
items in her cart. Her 5 packs of bacon cost $10 in total and she has 6 packets of chicken which each cost
twice as much as a pack of bacon. She also has 3 packs of strawberries, priced at $4 each, and 7 packs of
apples, each priced at half the price of a pack of strawberries. If Kelly's budget is $65 then how much money,
in dollars, does she have left in her budget?"""
    budget = 65
    bacon_packs = 5
    bacon_total_cost = 10
    chicken_packs = 6
    chicken_cost = 2 * bacon_total_cost
    strawberry_packs = 3
    strawberry_cost = 4
    apple_packs = 7
    apple_cost = strawberry_cost / 2
    total_cost = bacon_cost + chicken_cost + strawberry_cost + apple_cost
    money_left = budget - total_cost
    result = money_left
    return result

# Check each semantically complete block of the code to check for any logical reasoning errors. Logical
reasoning errors may include errors in the mathematical calculations, errors in the order of the steps, or
errors in the assumptions made. State the assumptions you made clearly. Ignore all the other types of errors.

# Let us go through the code step-by-step
    budget = 65
# looks good

# Let's check other parts
    bacon_packs = 5
    bacon_total_cost = 10
# looks good

# Let's check other parts
    chicken_packs = 6
    chicken_cost = 2 * bacon_total_cost
# wrong! according to the context, the cost of each packet of chicken is twice the cost of 1 packet of bacon.
We should use bacon_cost in place of bacon_total_cost to calculate the chicken pack cost correctly. Let's fix
it.

# Let's check other parts
    strawberry_packs
    strawberry_cost = 4
# looks good

# Let's check other parts
    apple_packs = 7
    apple_cost = strawberry_cost / 2
# looks good

# Let's check other parts
    total_cost = bacon_cost + chicken_cost + strawberry_cost + apple_cost
    money_left = budget - total_cost
    result = money_left
    return result
# wrong! we want to calculate the total cost of buying all the items so we should use the total cost of each
item instead of cost of 1 pack of each item. Let's fix it.

### END ###

...
```

Figure 5: Commonsense Feedback for Mathematical Reasoning

```
def solution():
    """Kelly is grocery shopping at a supermarket and is making sure she has enough in her budget for the
items in her cart. Her 5 packs of bacon cost $10 in total and she has 6 packets of chicken which each cost
twice as much as a pack of bacon. She also has 3 packs of strawberries, priced at $4 each, and 7 packs of
apples, each priced at half the price of a pack of strawberries. If Kelly's budget is $65 then how much money,
in dollars, does she have left in her budget?"""
    budget = 65
    bacon_packs = 5
    bacon_total_cost = 10
    chicken_packs = 6
    chicken_cost = 2 * bacon_cost
    strawberry_packs = 3
    strawberry_cost = 4
    apple_packs = 7
    apple_cost = strawberry_cost / 2
    total_cost = bacon_cost + chicken_cost + strawberry_cost + apple_cost
    money_left = budget - total_cost
    result = money_left
    return result

# Check each semantically complete block of code for any missing steps and suggest the correct way to add
them. Ignore all the other types of errors.

# Let us go through the code step-by-step
    budget = 65
# looks good

# Let's check other parts
    bacon_packs = 5
    bacon_total_cost = 10
# looks good

# Let's check other parts
    chicken_packs = 6
    chicken_cost = 2 * bacon_cost
# wrong! bacon_cost is missing. Let's add it.
# wrong! we need the total cost of chicken to calculate remaining budget. Let's add it.

# Let's check other parts
    strawberry_packs
    strawberry_cost = 4
# wrong! we need the total cost of strawberries to calculate remaining budget. Let's add it.

# Let's check other parts
    apple_packs = 7
    apple_cost = strawberry_cost / 2
# wrong! we need the total cost of apples to calculate remaining budget. Let's add it.

# Let's check other parts
    total_cost = bacon_cost + chicken_cost + strawberry_cost + apple_cost
    money_left = budget - total_cost
    result = money_left
    return result
# looks good

### END ###

...
```

Figure 6: Missing Step Feedback for Mathematical Reasoning

```
def solution():
    """Albert is wondering how much pizza he can eat in one day. He buys 2 large pizzas and 2 small pizzas. A
large pizza has 16 slices and a small pizza has 8 slices. Albert also has 5 pieces of cake. If he eats it all,
how many pieces does he eat that day?"""
    large_pizzas = 2
    small_pizzas = 2
    large_pizza_slices = 16
    small_pizza_slices = 8
    cake_pieces = 5
    total_slices = (large_pizzas * large_pizza_slices) + (small_pizzas * small_pizza_slices) + cake_pieces
    return total_slices

# Check each semantically complete block of code for any redundancy errors and suggest fixes. Redundancy
errors are steps that contain redundant information, which even though might be factual, is not required to
answer the question. Ignore all other types of errors.

# Let's go through the code step-by-step
    large_pizzas = 2
    small_pizzas = 2
# looks good

# Let's check other parts
    large_pizza_slices = 16
    small_pizza_slices = 8
# looks good

# Let's check other parts
    cake_pieces = 5
# wrong! while this information is accurate, we do not need this to calculate the answer. Let's remove it

# Let's check other parts
    total_slices = (large_pizzas * large_pizza_slices) + (small_pizzas * small_pizza_slices) + cake_pieces
# wrong! we should not add cake_pieces because we want to calculate the number of pizza pieces he can eat.
Let's fix it

### END ###

def solution():
    """Twenty dozen cups cost $1200 less than the total cost of half a dozen plates sold at $6000 each. Two
dozen party hats cost $400 less than the cost of half a dozen plates. Calculate the total cost of buying each
cup."""
    # logical reasoning
    plates = 6
    plate_cost = 6000
    plate_total_cost = plate_cost * plates
    party_hats = 12 * 2
    party_hats_total_cost = plate_total_cost - 400
    party_hat_cost = party_hats_total_cost / party_hats
    cups = 12 * 20
    cup_total_cost = plate_total_cost - 1200
    cup_cost = cup_total_cost / cups
    result = cup_cost
    return result

# Check each semantically complete block of code for any redundancy errors and suggest fixes. Redundancy
errors are steps that contain redundant information, which even though might be factual, is not required to
answer the question. Ignore all other types of errors.

# Let's go through the code step-by-step
    plates = 6
    plate_cost = 6000
# looks good

...
```

Figure 7: Redundancy Feedback for Mathematical Reasoning

```
...

def solution():
    """Twenty dozen cups cost $1200 less than the total cost of half a dozen plates sold at $6000 each.
Calculate the total cost of buying each cup."""
    plates = 6
    plate_cost = 6000
    cups = 12 * 20
    cup_cost = plate_cost
    result = cup_cost
    return result

# Check each semantically complete block of code and identify the variables that are not named correctly or
may cause confusion and fix the issues. State the assumptions you made when renaming the variables clearly.
Ignore all the other type of errors.

# Let us go through each variable step-by-step
    plates = 6
    plate_cost = 6000
# looks good

# Let's check other parts
    cups = 12 * 20
    cup_cost = plate_cost
# rename the variable cup_cost to cups_total_cost because we are given information about total cost of cups.
    cups = 12 * 20
    cups_total_cost = plate_cost

# Let's check other parts
    result = cup_cost
    return result
# looks good

# Okay, here is the rewrite:

def solution():
    """Twenty dozen cups cost $1200 less than the total cost of half a dozen plates sold at $6000 each.
Calculate the total cost of buying each cup."""
    plates = 6
    plate_cost = 6000
    cups = 12 * 20
    cups_total_cost = plate_cost
    result = cup_cost
    return result

### END ###

def solution():
    """Carly had 80 cards, 2/5 of the cards had the letter A on them, 1/2 of the remaining had the letter B,
5/8 of the rest had the letter C on them, and the others had the letter D. How many of the cards had the
letter D on them?"""
    cards_initial = 80
    cards_a = cards_initial * 2 / 5
    cards_left = cards_initial - cards_a
    cards_b = cards_left * 1 / 2
    cards_left = cards_left - cards_b
    cards_c = cards_left * 5 / 8
    cards_d = cards_left * 3 / 8
    result = cards_d
    return result

# Check each semantically complete block of code and identify the variables that are not named correctly or
may cause confusion and fix the issues. State the assumptions you made when renaming the variables clearly.
Ignore all the other type of errors.

...
```

Figure 8: Variable Naming Feedback for Mathematical Reasoning

```
...

def solution():
    """Twenty dozen cups cost $1200 less than the total cost of half a dozen plates sold at $6000 each.
Calculate the total cost of buying each cup."""
    plates = 0.5
    plate_cost = 6000
    cups = 12 * 20
    cups_total_cost = plate_cost
    result = cup_cost
    return result

Commonsense Feedback:
# Let us go through the code step-by-step
    plates = 0.5
    plate_cost = 6000
# wrong! The number of plates is wrong. The question mentions half a dozen plates. A dozen is 12. So half a
dozen is 6. Let's fix it.

# Let's check other parts
    cups = 12 * 20
    cup_total_cost = plate_cost
# wrong! the questions mentions total cost of all cups is 1200 less than total cost of plates. So we should
use plate_total_cost instead of plate_cost and subtract 1200 from it. Let's fix it.

Missing Step Feedback:
# Let us go through the code step-by-step
    plates = 0.5
    plate_cost = 6000
# wrong! we need to calculate the total cost of plates to calculate the cost of each cup. Let's add it.

# Let's check other parts
    result = cup_cost
    return result
# wrong! we need to calculate the cost of each cup to answer the question. Let's add it.

# Given the feedback and the original code, let's rewrite the code to incorporate all of the feedback. Don't
change anything unless it is mentioned in the feedback.

Okay, here is the rewrite:

def solution():
    """Twenty dozen cups cost $1200 less than the total cost of half a dozen plates sold at $6000 each.
Calculate the total cost of buying each cup."""
    # logical reasoning
    plates = 6
    plate_cost = 6000
    # missing step
    plate_total_cost = plate_cost * plates
    cups = 12 * 20
    # logical reasoning
    cup_total_cost = plate_total_cost - 1200
    # missing step
    cup_cost = cup_total_cost / cups
    result = cup_cost
    return result

...
```

Figure 9: Our Iterative Refinement prompt for Mathematical Reasoning

```
...

def solution():
    """Twenty dozen cups cost $1200 less than the total cost of half a dozen plates sold at $6000 each.
Calculate the total cost of buying each cup."""
    plates = 6
    plate_cost = 6000
    cups = 12 * 20
    cup_cost = plate_cost
    result = cup_cost
    return result

# There is an error in the code above because of lack of understanding of the question. What is the error? To
find the error, go through semantically complete blocks of the code, and check if everything looks good.

# Let us go through the error and check step-by-step
    plates = 6
    plate_cost = 6000
# looks good

# Let's check the other parts
    cups = 12 * 20
    cup_cost = plate_cost
# wrong! The cost of a cup is not the same as the cost of a plate. The cost of a cup is $1200 less than the
total cost of half a dozen plates sold at $6000 each. So we need to calculate the cost of a cup first (total
cost of half a dozen plates sold at $6000 each - $1200) and use that.

Okay! Here is the rewrite:

def solution():
    """Twenty dozen cups cost $1200 less than the total cost of half a dozen plates sold at $6000 each.
Calculate the total cost of buying each cup."""
    plates = 6
    plate_cost = 6000
    cups = 12 * 20
    total_cup_cost = (plate_cost * plates) - 1200
    cost_per_cup = total_cup_cost / cups
    return cost_per_cup

### END ###

def solution():
    """Carly had 80 cards, 2/5 of the cards had the letter A on them, 1/2 of the remaining had the letter B,
5/8 of the rest had the letter C on them, and the others had the letter D. How many of the cards had the
letter D on them?"""
    cards_initial = 80
    cards_a = cards_initial * 2 / 5
    cards_left = cards_initial - cards_a
    cards_b = cards_left * 1 / 2
    cards_left = cards_left - cards_b
    cards_c = cards_left * 5 / 8
    cards_d = cards_left * 3 / 8
    result = cards_d
    return result

# There is an error in the code above because of lack of understanding of the question. What is the error? To
find the error, go through semantically complete blocks of the code, and check if everything looks good.

# Let us go through the error and check step-by-step
    cards_initial = 80
# looks good

...
```

Figure 10: Iterative Refinement prompt from Self-Refine for Mathematical Reasoning

Passage: The French king, John II, had been held captive in England. The Treaty of Brétigny set his ransom at 3 million crowns and allowed for hostages to be held in lieu of John. The hostages included two of his sons, several princes and nobles, four inhabitants of Paris, and two citizens from each of the nineteen principal towns of France. While these hostages were held, John returned to France to try and raise funds to pay the ransom. In 1362 John's son Louis of Anjou, a hostage in English-held Calais, escaped captivity. So, with his stand-in hostage gone, John felt honor-bound to return to captivity in England. The French crown had been at odds with Navarre  since 1354, and in 1363 the Navarrese used the captivity of John II in London and the political weakness of the Dauphin to try to seize power. Although there was no formal treaty, Edward III supported the Navarrese moves, particularly as there was a prospect that he might gain control over the northern and western provinces as a consequence. With this in mind, Edward deliberately slowed the peace negotiations. In 1364, John II died in London, while still in honourable captivity. Charles V succeeded him as king of France. On 7 May 1364, one month after the dauphin's accession and three days before his coronation as Charles V, the Navarrese suffered a crushing defeat at the Battle of Cocherel.

Question: How many groups were held hostage in lieu of John II?

Answer: The Treaty of Brétigny set John II's ransom at 3 million crowns and allowed for hostages to be held in lieu of him. The hostages included two of his sons, several princes and nobles, four inhabitants of Paris, and two citizens from each of the nineteen principal towns of France. Thus, there were a total of twenty-four groups held hostage in lieu of John II. final_answer: 24

# Check each semantically complete block of the reasoning to check for any commonsense errors. Commonsense reasoning errors are errors about any relation or knowledge that is should be known from general world such as "all ducks are birds". State the assumptions you made clearly. Ignore all the other types of errors.

# Let's go line by line
    The Treaty of Brétigny set John II's ransom at 3 million crowns and allowed for hostages to be held in lieu of him.
# looks good.

# Let's check the next line
    The hostages included two of his sons, several princes and nobles, four inhabitants of Paris, and two citizens from each of the nineteen principal towns of France.
# looks good

# Let's check the next line
    Thus, there were a total of twenty-four groups held hostage in lieu of John II.
# wrong! This answer counts each person held hostage as a group, which doesn't make sense, since a group generally isn't a single person. The different groups that are held hostage in lieu of John II are his sons, princes and nobles, inhabitants of Paris, and citizens from each of the nineteen principal towns of France. Thus, there are four groups held hostage in lieu of John II.

### END ###

Passage: As of the census of 2000, there were 218,590 people, 79,667 households, and 60,387 families residing in the county.  The population density was 496 people per square mile (192/km\u00b2). There were 83,146 housing units at an average density of 189 per square mile (73/km\u00b2). The racial makeup of the county was 86.77% Race (United States Census), 9.27% Race (United States Census), 0.23% Race (United States Census), 1.52% Race (United States Census), 0.06% Race (United States Census), 0.69% from Race (United States Census), and 1.47% from two or more races.  1.91% of the population were Race (United States Census) or Race (United States Census) of any race. 22.5% were of German people, 13.1% Irish people, 9.8% Italian people, 9.2% English, 8.1% \"American\" and 6.0% Polish ancestry.

Question: How many more housing units are there than families?

Answer: The passage states that there were 79,667 households and 60,387 families residing in the county. 79,667 households minus 60,387 families equals 19,280. Thus, there are 19,280 more housing units than families. final_answer: 19,280

# Check each semantically complete block of the reasoning to check for any commonsense errors. Commonsense reasoning errors are errors about any relation or knowledge that is should be known from general world such as "all ducks are birds". State the assumptions you made clearly. Ignore all the other types of errors.

...

Figure 11: Commonsense Feedback for Question Answering

Passage: The French king, John II, had been held captive in England. The Treaty of Brétigny set his ransom at 3-million-crowns and allowed for hostages to be held in lieu of John. The hostages included two of his sons, several princes and nobles, four inhabitants of Paris, and two citizens from each of the nineteen principal towns of France. While these hostages were held, John returned to France to try and raise funds to pay the ransom. In 1362 John's son Louis of Anjou, a hostage in English-held Calais, escaped captivity. So, with his stand-in hostage gone, John felt honor-bound to return to captivity in England. The French crown had been at odds with Navarre since 1354, and in 1363 the Navarrese used the captivity of John II in London and the political weakness of the Dauphin to try to seize power. Although there was no formal treaty, Edward III supported the Navarrese moves, particularly as there was a prospect that he might gain control over the northern and western provinces as a consequence. With this in mind, Edward deliberately slowed the peace negotiations. In 1364, John II died in London, while still in honourable captivity. Charles V succeeded him as king of France. On 7 May 1364, one month after the dauphin's accession and three days before his coronation as Charles V, the Navarrese suffered a crushing defeat at the Battle of Cocherel.

Question: How many years before Navarrase used the captivity of John II?

Answer: The French crown had been at odds with Navarre since 1354. Thus, Navarrase used the captivity of John II 10 years later in 1364. final_answer: 10

# Check each semantically complete block of reasoning for any infactual information and remove or fix it. Any information not present in the context is considered infactual. Ignore all other types of errors.

# Let's go line by line
    The French crown had been at odds with Navarre since 1354.
# looks good

# Let's check the next line
    Thus, Navarrase used the captivity of John II 10 years later in 1364.
# wrong! From the passage, "in 1363 the Navarrese used the captivity of John II in London". This line falsely states that the Navarrese used the captivity of John II in 1364 when they actually used it in 1363. Let's fix it

### END ###

Passage: To start the season, the Lions traveled south to Tampa, Florida to take on the Tampa Bay Buccaneers. The Lions scored first in the first quarter with a 23-yard field goal by Jason Hanson. The Buccaneers tied it up with a 38-yard field goal by Connor Barth, then took the lead when Aqib Talib intercepted a pass from Matthew Stafford and ran it in 28 yards. The Lions responded with a 28-yard field goal. In the second quarter, Detroit took the lead with a 36-yard touchdown catch by Calvin Johnson, and later added more points when Tony Scheffler caught an 11-yard TD pass. Tampa Bay responded with a 31-yard field goal just before halftime. The second half was relatively quiet, with each team only scoring one touchdown. First, Detroit's Calvin Johnson caught a 1-yard pass in the third quarter. The game's final points came when Mike Williams of Tampa Bay caught a 5-yard pass.  The Lions won their regular season opener for the first time since 2007

Question: How many field goals did the Lions score?

Answer: The first was a 23-yard field goal by Jason Hanson in the first quarter. The second was a 28-yard field goal in response to the Buccaneers' interception. And the third was a 36-yard field goal in the second quarter. So they scored three field goals. final_answer: 3

# Check each semantically complete block of reasoning for any infactual information and remove or fix it. Any information not present in the context is considered infactual. Ignore all other types of errors.

# Let's go line by line
    The first was a 23-yard field goal by Jaon Hanson in the first quarter.
# looks good

# Let's check the next line
    The second was a 28-yard field goal in response to the Buccaneers' interception.
# looks good

# Let's check next line
    And the third was a 36-yard field goal in the second quarter.
# wrong! This answer count the 36-yard touchdown catch by Calvin Johnson in second quarter as a field goal. This is wrong because a touchdown is not a field goal. Thus, this should not be counted in the answer.
...

Figure 12: Factuality Feedback for Question Answering

Passage: To start the season, the Lions traveled south to Tampa, Florida to take on the Tampa Bay Buccaneers. The Lions scored first in the first quarter with a 23-yard field goal by Jason Hanson. The Buccaneers tied it up with a 38-yard field goal by Connor Barth, then took the lead when Aqib Talib intercepted a pass from Matthew Stafford and ran it in 28 yards. The Lions responded with a 28-yard field goal. In the second quarter, Detroit took the lead with a 36-yard touchdown catch by Calvin Johnson, and later added more points when Tony Scheffler caught an 11-yard TD pass. Tampa Bay responded with a 31-yard field goal just before halftime. The second half was relatively quiet, with each team only scoring one touchdown. First, Detroit's Calvin Johnson caught a 1-yard pass in the third quarter. The game's final points came when Mike Williams of Tampa Bay caught a 5-yard pass. The Lions won their regular season opener for the first time since 2007.

Question: Who caught the shortest touchdown pass?

Answer: The shortest touchdown pass was caught by Tony Scheffler. He caught an 11-yard TD pass. final_answer: Tony Scheffler

# Check each semantically complete block of reasoning for any missing steps and suggest the correct way to add them. Ignore all the other types of errors.

# Let's check the answer step-by-step
    The shortest touchdown pass was caught by Tony Scheffler. He caught an 11-yard TD pass. final_answer: Tony Scheffler
# wrong! This answer missed the 1-yard pass caught by Calvin Johnson and 5-yard pass caught by Mike Williams. So the answer should be Let's add this information to the answer.

### END ###

Passage: The French king, John II, had been held captive in England. The Treaty of Br\u00e9tigny set his ransom at 3\u00a0million\u00a0crowns and allowed for hostages to be held in lieu of John. The hostages included two of his sons, several princes and nobles, four inhabitants of Paris, and two citizens from each of the nineteen principal towns of France. While these hostages were held, John returned to France to try and raise funds to pay the ransom. In 1362 John's son Louis of Anjou, a hostage in English-held Calais, escaped captivity. So, with his stand-in hostage gone, John felt honor-bound to return to captivity in England. The French crown had been at odds with Navarre since 1354, and in 1363 the Navarrese used the captivity of John II in London and the political weakness of the Dauphin to try to seize power. Although there was no formal treaty, Edward III supported the Navarrese moves, particularly as there was a prospect that he might gain control over the northern and western provinces as a consequence. With this in mind, Edward deliberately slowed the peace negotiations. In 1364, John II died in London, while still in honourable captivity. Charles V succeeded him as king of France. On 7 May 1364, one month after the dauphin's accession and three days before his coronation as Charles V, the Navarrese suffered a crushing defeat at the Battle of Cocherel.

Question: At least how many people were held hostage in place of John II?

Answer: The Treaty of Br\u00e9tigny set John II's ransom at 3 million crowns and allowed for hostages to be held in lieu of John. The hostages included two of his sons, several princes and nobles, four inhabitants of Paris, and two citizens from each of the nineteen principal towns of France. Thus at least 2 + 19 * 2 = 40 people were held hostage in place of John II. final_answer: 40

# Check each semantically complete block of reasoning for any missing steps and suggest the correct way to add them. Ignore all the other types of errors.

# Let's check the answer step-by-step
   The Treaty of Br\u00e9tigny set John II's ransom at 3 million crowns and allowed for hostages to be held in lieu of John.
# looks good

# Let's check other parts
    The hostages included two of his sons, several princes and nobles, four inhabitants of Paris, and two citizens from each of the nineteen principal towns of France.
# looks good

# Let's check other parts
    Thus at least 2 + 19 * 2 = 40 people were held hostage in place of John II. final_answer: 44
# wrong! this step missed counting the 4 inhabitants of Paris. So the answer should be changed. So the answer will be 40 + 4 = 44. Let's add this.

...

Figure 13: Missing Step Feedback for Question Answering

Passage: The French king, John II, had been held captive in England. The Treaty of Brétigny set his ransom at 3 million crowns and allowed for hostages to be held in lieu of John. The hostages included two of his sons, several princes and nobles, four inhabitants of Paris, and two citizens from each of the nineteen principal towns of France. While these hostages were held, John returned to France to try and raise funds to pay the ransom. In 1362 John's son Louis of Anjou, a hostage in English-held Calais, escaped captivity. So, with his stand-in hostage gone, John felt honor-bound to return to captivity in England. The French crown had been at odds with Navarre since 1354, and in 1363 the Navarrese used the captivity of John II in London and the political weakness of the Dauphin to try to seize power. Although there was no formal treaty, Edward III supported the Navarrese moves, particularly as there was a prospect that he might gain control over the northern and western provinces as a consequence. With this in mind, Edward deliberately slowed the peace negotiations. In 1364, John II died in London, while still in honourable captivity. Charles V succeeded him as king of France. On 7 May 1364, one month after the dauphin's accession and three days before his coronation as Charles V, the Navarrese suffered a crushing defeat at the Battle of Cocherel.

Question: How many groups were held hostage in lieu of John II?

Answer: The Treaty of Brétigny set John II's ransom at 3 million crowns and allowed for hostages to be held in lieu of him. The hostages included two of his sons, several princes and nobles, four inhabitants of Paris, and two citizens from each of the nineteen principal towns of France. Thus, there were a total of twenty-four groups held hostage in lieu of John II. final_answer: 24

# Check each semantically complete block of reasoning for any redundancy errors and suggest fixes. Redundancy errors are steps that contain redundant information, which even though might be factual, is not required to answer the question. Ignore all other types of errors.

# Let's go line by line
    The Treaty of Brétigny set John II's ransom at 3 million crowns and allowed for hostages to be held in lieu of him.
# wrong! The information that John II's ransom was set at 3 million crowns is redundant and and not relevant to the question of how many groups were held hostage in lieu of him. Let's remove that information.

# Let's check the next line
    The hostages included two of his sons, several princes and nobles, four inhabitants of Paris, and two citizens from each of the nineteen principal towns of France.
# wrong! The exact number of hostages in each group is not relevant to the question of how many groups were held hostage in lieu of John II. Let's remove that information.

# Let's check the next line
    Thus, there were a total of twenty-four groups held hostage in lieu of John II.
# looks good

### END ###

Passage: Trying to snap a two-game skid, the Bills flew to Gillette Stadium for a Week 3 divisional fight with the New England Patriots. In the first quarter, QB J. P. Losman was immediately injured on the first offensive play of the game. He would finish the series, but ended up on the bench for the rest of the game. After New England took the lead with kicker Stephen Gostkowski's 24-yard field goal, rookie QB Trent Edwards played the rest of the game for Buffalo. The Bills would get their only score of the game as RB Marshawn Lynch got an 8-yard TD run, and a Rian Lindell extra point put the Bills ahead surprisingly 7-3. However, in the second quarter, the Patriots were able to open up their running game when Bills rookie standout Paul Posluszny was lost due to a broken arm. This left passing lanes open, and for the rest of the game, the Patriots dominated. QB Tom Brady's 8-yard TD pass to TE Benjamin Watson and a 3-yard TD pass to WR Randy Moss made it 17-7 at the half. In the third quarter, New England continued its conquest with Brady's 4-yard TD pass to WR Jabar Gaffney and RB Sammy Morris' 4-yard TD run. In the fourth quarter, the Patriots ended the day with Brady and Moss hooking up with each other again on a 45-yard TD pass.

Question: How many points were scored by Patriots in the first half?

Answer: In the first half, the Bills scored 7 points with a Marshawn Lynch 8-yard TD run and a Rian Lindell extra point. The Patriots scored 17 points with a Stephen Gostkowski 24-yard field goal, a Tom Brady 8-yard TD pass to TE Benjamin Watson, and a 3-yard TD pass to WR Randy Moss. Thus, a total of 24 points were scored in the first half. final_answer: 24

...

Figure 14: Redundancy Feedback for Question Answering

```
Passage: The French king, John II, had been held captive in England. The Treaty of Brétigny set his ransom at
3 million crowns and allowed for hostages to be held in lieu of John. The hostages included two of his sons,
several princes and nobles, four inhabitants of Paris, and two citizens from each of the nineteen principal
towns of France. While these hostages were held, John returned to France to try and raise funds to pay the
ransom. In 1362 John's son Louis of Anjou, a hostage in English-held Calais, escaped captivity. So, with his
stand-in hostage gone, John felt honor-bound to return to captivity in England. The French crown had been at
odds with Navarre  since 1354, and in 1363 the Navarrese used the captivity of John II in London and the
political weakness of the Dauphin to try to seize power. Although there was no formal treaty, Edward III
supported the Navarrese moves, particularly as there was a prospect that he might gain control over the
northern and western provinces as a consequence. With this in mind, Edward deliberately slowed the peace
negotiations. In 1364, John II died in London, while still in honourable captivity. Charles V succeeded him as
king of France. On 7 May 1364, one month after the dauphin's accession and three days before his coronation as
Charles V, the Navarrese suffered a crushing defeat at the Battle of Cocherel.

Question: How many groups were held hostage in lieu of John II?

Answer: The Treaty of Brétigny set John II's ransom at 3 million crowns and allowed for hostages to be held in
lieu of him. The hostages included two of his sons, several princes and nobles, four inhabitants of Paris, and
two citizens from each of the nineteen principal towns of France. Thus, there were a total of twenty-four
groups held hostage in lieu of John II. final_answer: 24

# There is a error in the answer above. What is the error? To find the error go through the answer line by
line and check that everything looks good.

# Let's go line by line
    The Treaty of Brétigny set John II's ransom at 3 million crowns and allowed for hostages to be held in
lieu of him.
# wrong! this information is not needed to answer the question. Let's remove it.

# Let's check the next line
    The hostages included two of his sons, several princes and nobles, four inhabitants of Paris, and two
citizens from each of the nineteen principal towns of France.
# looks good

# Let's check the next line
    Thus, there were a total of twenty-four groups held hostage in lieu of John II.
# wrong! This answer counts each person held hostage as a group, which doesn't make sense, since a group
generally isn't a single person. The different groups that are held hostage in lieu of John II are his sons,
princes and nobles, inhabitants of Paris, and citizens from each of the nineteen principal towns of France.
Thus, there are four groups held hostage in lieu of John II.

### END ###

Passage: To start the season, the Lions traveled south to Tampa, Florida to take on the Tampa Bay Buccaneers.
The Lions scored first in the first quarter with a 23-yard field goal by Jason Hanson. The Buccaneers tied it
up with a 38-yard field goal by Connor Barth, then took the lead when Aqib Talib intercepted a pass from
Matthew Stafford and ran it in 28 yards. The Lions responded with a 28-yard field goal. In the second quarter,
Detroit took the lead with a 36-yard touchdown catch by Calvin Johnson, and later added more points when Tony
Scheffler caught an 11-yard TD pass. Tampa Bay responded with a 31-yard field goal just before halftime. The
second half was relatively quiet, with each team only scoring one touchdown. First, Detroit's Calvin Johnson
caught a 1-yard pass in the third quarter. The game's final points came when Mike Williams of Tampa Bay caught
a 5-yard pass.  The Lions won their regular season opener for the first time since 2007

Question: How many field goals did the Lions score?

Answer: The first was a 23-yard field goal by Jason Hanson in the first quarter. The second was a 28-yard
field goal in response to the Buccaneers' interception. And the third was a 36-yard field goal in the second
quarter. So they scored three field goals. final_answer: 3

# There is a error in the answer above. What is the error? To find the error go through the answer line by
line and check that everything looks good.

# Let's go line by line
    The first was a 23-yard field goal by Jaon Hanson in the first quarter.
# looks good
...
```

Figure 15: Self-Refine style Iterative Refinement prompt for Question Answering

Passage: To start the season, the Lions traveled south to Tampa, Florida to take on the Tampa Bay Buccaneers. The Lions scored first in the first quarter with a 23-yard field goal by Jason Hanson. The Buccaneers tied it up with a 38-yard field goal by Connor Barth, then took the lead when Aqib Talib intercepted a pass from Matthew Stafford and ran it in 28 yards. The Lions responded with a 28-yard field goal. In the second quarter, Detroit took the lead with a 36-yard touchdown catch by Calvin Johnson, and later added more points when Tony Scheffler caught an 11-yard TD pass. Tampa Bay responded with a 31-yard field goal just before halftime. The second half was relatively quiet, with each team only scoring one touchdown. First, Detroit's Calvin Johnson caught a 1-yard pass in the third quarter. The game's final points came when Mike Williams of Tampa Bay caught a 5-yard pass. The Lions won their regular season opener for the first time since 2007.

Question: How many points did the buccaneers need to tie in the first?

Answer: The Buccaneers tied the score in the first quarter with "a 38-yard field goal by Connor Barth". A field goal is worth 3 points in American football. Thus, the bucaneers needed 3 points to tie the score in the first quarter. final_answer: 3

### END ###

Passage: Peck also received Oscar nominations for his roles in The Keys of the Kingdom (film) (1944), The Yearling (film) (1946), Gentlemans Agreement (1947) and Twelve OClock High (1949). Other notable films in which he appeared include Spellbound (1945 film) (1945), The Gunfighter (1950), Roman Holiday (1953), Moby Dick (1956 film) (1956, and its Moby Dick (1998 miniseries)), The Big Country (1958), The Bravados (1958), Pork Chop Hill (1959), The Guns of Navarone (film) (1961), Cape Fear (1962 film) (1962, and its Cape Fear (1991 film)), How the West Was Won (film) (1962), The Omen (1976) and The Boys from Brazil (film) (1978).

Question: What film did Peck receive an Oscar nomination first: The Keys of the Kingdom or The Yearling?

Answer: Peck received an Oscar nomination for The Keys of the Kingdom in 1944 and the Yearling in 1946. 1944 is before 1946, so Peck received an Oscar nomination for The Keys of the Kingdom first. final_answer: "The Keys of the Kingdom"

### END ###

Passage: The Spanish were able to complete their escape to Tlaxcala. There, they were given assistance, since all 440 of them were wounded, with only 20 horses left. Maxixcatzin, Xicotencatl the Elder and Chichimecatecle told Cortés's men: "Consider yourselves at home. Rest...do not think it a small thing that you have escaped with your lives from that strong city...if we thought of you as brave men before, we consider you much braver now.":306-07 Cortés got reinforcements when the Panuco River settlement was abandoned, and supply ships arrived from Cuba and Spain. Cortés also had built 13 brigantines then had them mounted with cannons, turning Lake Texcoco into a strategic body of water to assault Tenochtitlan. Xicotencatl the Younger, however, sought an alliance with the Mexicans, but was opposed.:309-11 Cortés sent Diego de Ordaz, and the remnants of Navarez's men, on a ship to Spain, and Alonso de Ávila on a ship to Santo Domingo to represent his case in the Royal Courts.:311 Cortés was able to pacify the country, after the indigenous realized the Spaniards put "an end to the rape and robbery that the Mexicans practised." Finally, Xicotencatl the Elder, baptized as Don Lorenzo de Vargas, agreed to support Cortés's expedition against Texcoco. According to Bernal Diaz, he sent more than ten thousand warriors under the command of Chichimecatecle as Cortés marched on the day after Christmas 1520.:309, 311-12

Question: On what date did 10,000 warriors march under Chichimecatecle?

Answer: According to Bernal Diaz, 10,000 warriors marched under Chichimecatecle on "the day after Christmas 1520". The day after christmas is 26 December. Thus, 10,000 warriors marched under Chichimecatecle on 26 December 1520. final_answer: 26-December-1520

### END ###

Use the following passage to answer the questions below it with either a number, date, or span from the passage. Format your answer with "final_answer: " and then your final answer. If your answer is a number, give just the numeric answer. If your answer is a date, then give the day in numbers, the month by its name, and the year in numbers. If your answer is a span, surround it in quotes.

Figure 16: Initialization prompt for Question Answering