# OpenReview forum: "MAF: Multi-Aspect Feedback for Improving Reasoning in Large Language Models"
_EMNLP/2023/Conference — EMNLP 2023 Main_

### Official Review · Reviewer_J3st · 2023-07-25

**Soundness:** 3

**Excitement:**

3: Ambivalent: It has merits (e.g., it reports state-of-the-art results, the idea is nice), but there are key weaknesses (e.g., it describes incremental work), and it can significantly benefit from another round of revision. However, I won't object to accepting it if my co-reviewers champion it.

**Paper Topic And Main Contributions:**

Given that existing approaches rely on single generic feedback and fail to address the diverse error types in LLM reasoning, this paper proposes a Multi-Aspect Feedback framework based on multiple feedback modules, including LMs and external tools. Overall, this paper might contribute to the "NLP engineering experiment".

**Questions For The Authors:**

Question A: How did you choose the order of feedback type in the experiment? Do you think different orders of feedback will have impacts on model performance? Like doing the missing step first, and then redundancy, or the reverse order?

**Reasons To Accept:**

- Overall experimental results show the effectiveness of the proposed framework in several reasoning tasks.
- The proposed framework is modular and the modules are plug-and-play, which provides good flexibility for using new models and tools in new tasks.
- This paper found that over-refinement can be a problem in iterative refinement frameworks, which might be interesting and useful to the community.

**Reasons To Reject:**

- In the contributions, the authors claim their proposed framework outperforms baselines on Math reasoning, such as GSM8K and GSM-IC, but in Sec 4.1 they say that the performance on GSM8K has been saturated and use GSM-IC for the experiment (they indeed show the results of GSM-IC in the paper only).
- For the two proposed refinement strategies: Eager Refinement and Lazy Refinement, although the author gives an example to show that these two methods are ‘complementary’,  I personally think that a detailed comparison of them should be included in the experiments.
- In Table 3, the results show that some modules are not necessary, and even might hurt the framework according to the results of MAF (VN, Red, Com). In this regard, and as a long paper, the authors might need to provide more comprehensive experiments to show the effectiveness and necessity of each module for each task.

**Reproducibility:**

3: Could reproduce the results with some difficulty. The settings of parameters are underspecified or subjectively determined; the training/evaluation data are not widely available.

**Reviewer Confidence:**

4: Quite sure. I tried to check the important points carefully. It's unlikely, though conceivable, that I missed something that should affect my ratings.

**Typos Grammar Style And Presentation Improvements:**

- Table 1 is placed on page 3, but is not mentioned until page 7, and the table needs more clarification for easy reading.
- If I understand correctly, Table 3 shows the ablation study results for Sec 4.5, but that section doesn't mention it.
- In the related work, ‘prefixing the solution generation with Let’s think step-by-step’ is not proposed by Wei et al. 2023 but Kojima et al. 2023.
- The pictures in the appendix are very blurry and unreadable.

---

> ### Author Rebuttal · Authors · 2023-08-29
>
> Thank you for your comprehensive review and actionable feedback! We carefully consider your concerns and address them below.
>
> **3A. In the contributions, the authors claim their proposed framework outperforms baselines on Math reasoning, such as GSM8K and GSM-IC, but in Sec 4.1 they say that the performance on GSM8K has been saturated and use GSM-IC for the experiment (they indeed show the results of GSM-IC in the paper only).**
> > This is indeed a mistake from our side. In the contribution section, it should be “mathematical reasoning (GSM-IC)” rather than “mathematical reasoning (GSM8K and GSM-IC)“. As stated in our paper Lines 350 - 355, we did not choose GSM8K because the performance on GSM8K has been too saturated. Thanks for noticing this inconsistency.
> >
> > Nonetheless, we still conduct additional experiments to investigate the effects of MAF on GSM8k. We will add the results presented below to the revised version of the manuscript.
>
>
> | Model | GSM8K |
> |--------|-----------|
> | GPT3.5           | 69.20 |
> | \+ Self-Refine | 69.20 |
> | \+ Self-Refine (Oracle) | *77.40* |
> | \+ MAF | **69.80** |
> |\+ MAF (Oracle) | 73.40 |
> | ChatGPT | 71.80 |
> | \+ Self-Refine | **74.60** |
> | \+ Self-Refine (Oracle) | *79.40* |
> | \+ MAF | 73.20 |
> | \+ MAF (Oracle) | 76.60 |
>
> **3B. For the two proposed refinement strategies: Eager Refinement and Lazy Refinement, although the author gives an example to show that these two methods are ‘complementary’, I personally think that a detailed comparison of them should be included in the experiments.**
> > We agree that elaborating on the interplay between eager and lazy refinement modules would improve this work. Following suit, we conduct a new experiment on a subset of the GSM-IC dataset where we compare our proposed setting with all modules in either lazy or eager mode.
>
> | Model | GSM-IC 100 | GSM-IC 100 (Oracle) | Over-Refined Instances |
> |--------|-----------------|-------------------------|-----------------------------------------|
> | GPT-3.5 | **71** | 71 | NA |
> | \+ MAF | 67 | *85* | 13 |
> | \+ All Lazy | 66 | 81 | 10 |
> | \+ All Eager | 66 | *85* | 17 |
> | ChatGPT | 73 | 73 | NA |
> | \+ MAF | **78** | *81* | 3 |
> | \+ All Lazy | 69 | 80 | 9 |
> | \+ All Eager | 73 | 80 | 6 |
>
> > As can be seen from the results, our framework which combines eager and lazy refinement does indeed outperform all lazy feedback and all eager feedback across both the standard and Oracle verifier settings. Over-Refined Instances in the above table show the number of instances where the model was able to generate the correct solution at Time Step $i$, but regressed at some future time step. We can also see that using all feedback modules in either lazy or eager mode also exacerbates this problem.
> >
> > Logically, one reason that we do not combine all feedbacks in one lazy feedback is that different feedbacks may result in revisions that interfere with each other. For example, Variable Naming feedback may suggest renaming a variable while Missing Step feedback suggests adding a line that uses that variable, resulting in an incorrect final answer. Having all feedback modules be eager also has its weaknesses, as any errors caused by the last feedback module in the sequence will not have a chance to be corrected. Overall, different feedback modules may be better suited to either eager or lazy refinement, so using a mix of them maximizes performance.
>
> > There are also practical concerns that favor using a mix of eager and lazy feedback. Using only lazy feedback can result in many feedback categories being compiled into one prompt. Even with our feedback summarization technique, this can lead to the iteration prompt being longer than the context window of many widely available models. On the other hand, using only eager feedback will result in the solution being rewritten for every feedback module, which will result in high token usage.
> >
> > We will add these details and a new ablation study comparing Lazy and Eager refinement strategies in our revised manuscript.
>
> **3C. In Table 3, the results show that some modules are not necessary, and even might hurt the framework according to the results of MAF (VN, Red, Com). In this regard, and as a long paper, the authors might need to provide more comprehensive experiments to show the effectiveness and necessity of each module for each task.**
> > We agree with the reviewer that some modules have small to no effect on the final results. One of the potential reasons for this is the provided ablation study (removing one feedback module at a time) does not capture the interplay of different feedback modules together. However, due to limited time, we were not able to do a detailed ablation study at this time on the interplay between multiple feedback modules. We will add an ablation on the importance of different feedback modules on each task in our final version.
>
> **3D. How did you choose the order of feedback type in the experiment? Do you think different orders of feedback will have impacts on model performance? Like doing the missing step first, and then redundancy, or the reverse order?**
> > Thank you for raising an interesting and important question. We chose the order of the feedback types intuitively. However, we agree that this is an important question and should be answered in this work.
> > To investigate how the order of various feedback modules affects our model’s performance, we conduct a new ablation study where we try multiple orders for our feedback modules. Specifically, we reverse the order of the Eager Refine modules, and Lazy Refine Modules respectively. We also try reversing the execution order of Eager and Lazy modules, i.e. instead of first applying the eager feedback we first apply all the lazy feedback modules and then apply Eager modules.
>
> | Model | GSM-IC 100 | GSM-IC 100 (Oracle) |
> |--------|----------------|--------------------------|
> | ChatGPT | 73 | 73 |
> | + MAF [Reported] - VN, Syntax, (MS, Com, Red) | **78** | **81** |
> | + MAF - Syntax, VN, (MS, Com, Red) | 73 | 77 |
> | + MAF - VN, Syntax (Red, Com, MS) | 75 | 78 |
> | + MAF - (MS, Com, Red), VN, Syntax | 74 | 80 |
> > VN refers to Variable Naming, MS refers to Missing Step, Com refers to Commonsense, and Red refers to Redundancy. All the modules inside parentheses are Lazy modules and other modules use eager refine strategy.
>
> > As seen from the table above, reversing the order of Eager Refine modules does affect the performance. This is expected because eager modules by design generate refined solutions immediately and in this particular case, generating Variable Naming feedback after fixing syntax errors can introduce new syntax errors. Whereas, changing the order of Lazy feedback modules ideally should have minimal effect on the performance because the feedback from all the modules is passed as one long feedback to the model. However, we note that this is not the case and the model’s performance does depend on the order of lazy feedback modules. We posit that this may be due to the length of the feedback. We will provide a more detailed study on the order of Lazy feedback modules in our revised manuscript.
>
> > Furthermore, in the reported setting, we execute all Eager Refine feedback before moving on to lazy modules because of the nature of the eager feedback modules. The syntactic eager feedback modules (Variable Naming and Syntax) primarily aim to make the generated program valid Python code. We hypothesized that placing these two modules first would result in higher quality feedback from the semantic modules (redundancy, commonsense, missing step) because the variable names and correct syntax could help align the feedback generation with the original problem. This is supported experimentally by testing our MAF framework on a subset of GSM-IC problems with the feedback modules reversed, such that the lazy modules come first, then variable naming, then syntax, which resulted in decreased performance. We will add these details in the revised manuscript.
>
> We are also grateful for the reviewer’s feedback on the overall presentation of our manuscript and will revise the manuscript accordingly. We hope this response adequately addresses your queries. Thank you once more for your constructive feedback.

---

### Official Review · Reviewer_Bg9c · 2023-08-12

**Typos Grammar Style And Presentation Improvements:** 1. The feedback module named "calcula…
**Soundness:** 3

**Excitement:**

4: Strong: This paper deepens the understanding of some phenomenon or lowers the barriers to an existing research direction.

**Paper Topic And Main Contributions:**

This paper introduces a framework called Multi-Aspect Feedback (MAF) for improving reasoning in large language models. The framework decouples feedback generation for different error categories and uses error-specific tools to provide feedback and refine the solution. The paper demonstrates the efficacy of MAF on various reasoning tasks and compares it with other iterative refinement baselines. The experimental results show that MAF outperforms the base language models and Self-Refine on complex reasoning tasks.

**Questions For The Authors:**

1. The "redundancy" module is used in all three evaluated tasks. Does this module share the same prompt across different types of tasks?
2. How do you implement "code interpreter" module?

**Reasons To Accept:**

1. This paper improves self-Refine framework by decoupling feedback generation for different error categories, allowing for error-specific tools to be used.
2. The experimental results demonstrate the effectiveness of MAF in improving the performance of LLMs on complex reasoning tasks.

**Reasons To Reject:**

1. The proposed MAF pipeline only supports generating feedback for pre-defined error types, and it might miss other error types that are not covered.
2. Human evaluation and case studies should be included in the final version.

**Reproducibility:**

3: Could reproduce the results with some difficulty. The settings of parameters are underspecified or subjectively determined; the training/evaluation data are not widely available.

**Reviewer Confidence:**

4: Quite sure. I tried to check the important points carefully. It's unlikely, though conceivable, that I missed something that should affect my ratings.

---

> ### Author Rebuttal · Authors · 2023-08-29
>
> We thank the reviewer for their comprehensive review and insightful questions! We carefully address the questions and concerns below.
>
> **2A. The proposed MAF pipeline only supports generating feedback for pre-defined error types, and it might miss other error types that are not covered.**
> > We would like to emphasize that we have considered most of the error categories found in reasoning problems. However, for applying MAF to non-reasoning (Style Transfer, Summarization etc.) problems, we agree with the reviewer that our framework supports generating feedback only for pre-defined error types and it can be limiting in those cases.
> >
> > For those cases, We would like to assert that since our model supports plug-and-play style feedback modules, a new feedback module can be designed to target a task-specific error category. Moreover, our framework also supports using a fine-tuned model as a feedback module which provides even more freedom in fine-tuning smaller models for specific error categories that cannot be successfully addressed by in-context learning or external tool-based feedback modules.
> >
> > Overall, we believe that our framework presents a holistic approach to improving reasoning capabilities in LLMs by presenting a varied set of error categories that are often encountered in reasoning tasks and also provides extensibility by designing new task-specific feedback modules.
>
> **2B. Human evaluation and case studies should be included in the final version.**
> > Currently, we incorporate human evaluation into our metrics for the Entailment Bank dataset, where we asked human annotators to evaluate whether each entailment was valid or not. For GSM-IC and DROP, because the answers are objectively defined, we believe that using automatic evaluation is a sufficient measure of performance.
> >
> > If possible, we would like for the reviewer to clarify what additional areas they believe could be improved through human evaluation.
> >
> > In regards to case studies, we agree that qualitative examples of how MAF iteratively improves a generated solution would benefit our manuscript. We commit to adding such examples from our experiments in the appendix of our revised version.
>
> **2C. The "redundancy" module is used in all three evaluated tasks. Does this module share the same prompt across different types of tasks?**
> > In our work, all the in-context learning-based feedback modules are defined as a combination of a Task Instruction and Prompt Examples. When a feedback module for an error category is used across multiple tasks, the Task Instruction remains the same, however, the Prompt Examples are taken from the training set of the corresponding task.
> >
> >We provide the task instruction used for the Redundancy module below. This task instruction remains the same in all the tasks.
> >
> > ```Check each step/semantically complete block of code for any redundancy errors and suggest fixes. Redundancy errors are steps that contain redundant information, which even though might be factual, is not required to answer the question. Ignore all other types of errors.```
>
> **2D. How do you implement the "code interpreter" module?**
> > The generated code is passed as a string using Python Repl. This module requires the generated code to follow a specific format. For example, in our case we expect the model to generate a function named ‘solution’ which contains the correct logic. The stdout and stderr from the execution are then used as feedback. In a case, where the model generates no error, the feedback contains no actionable items and thus the generated code remains the same after refinement. If there is an error, we use that error message as a feedback for the model.
>
> **2E. The feedback module named "calculator" is illustrated in Figure 1, but it seems that this module is actually unimplemented (see Table 1).**
> > We would like to note that for GSM-IC since we are generating a Python program, we do not need a calculator separately. Since all the mathematical calculations are conducted as part of the program logic, there is no scope for calculation errors. The figure shows the calculator module separately for clarity and possibilities of different types of feedback modules.
>
> We are also grateful for the reviewer’s feedback on the overall presentation of our manuscript and will revise the manuscript accordingly.
> We hope this response adequately addresses your queries. Thank you once more for your constructive feedback.

---

### Official Review · Reviewer_zRzu · 2023-08-12

**Soundness:** 4

**Excitement:**

5: Transformative: This paper is likely to change its subfield or computational linguistics broadly. It should be considered for a best paper award. This paper changes the current understanding of some phenomenon, shows a widely held practice to be erroneous in someway, enables a promising direction of research for a (broad or narrow) topic, or creates an exciting new technique.

**Paper Topic And Main Contributions:**

This long paper entitled “MAF: Multi-Aspect Feedback for Improving Reasoning in Large Language Models” can contribute to NLP engineering. In this paper, authors consider not only self-refine feedback but also redundancy, repetition, and missing step feedback iteratively for self-improvement in mathematical reasoning, logical reasoning, and question answering. They found that improvements up to about 20% were achieved through their approach, MAF. Additionally, MAF seems to be useful and promising to any other tasks in NLP.

**Questions For The Authors:**

A. The authors took the categories from Golovneva et al. (2022) and made a new classification for defining feedback types. I wonder if the different types of feedback have different effects on the performance improvement. And, what additional kinds of feedback can be added to improve the performance of different tasks using LLMs?
B. What other NLP tasks can be benefited from MAF? And why do you think so?
C. Could you explain more about over-refinement problem and in what conditions the problem occurs and how your MAF can mitigate the problem?

**Reasons To Accept:**

This long paper can contribute to the improvement of reasoning using LLMs. MAF is a novel iterative refinement framework that decouples the feedback modules and takes advantage of error-specific tools to generate feedback. Its performance seems to be not only useful to mathematical reasoning, logical reasoning, and question answering that the authors examined, but also useful to other tasks in NLP. Additionally, the authors stated that overstatement is a problem. This problem can be widely discussed in the NLP community.

**Reasons To Reject:**

The authors may confirm if they considered all items in the responsible NLP checklist provided by ACL.

**Reproducibility:**

5: Could easily reproduce the results.

**Reviewer Confidence:**

4: Quite sure. I tried to check the important points carefully. It's unlikely, though conceivable, that I missed something that should affect my ratings.

---

> ### Author Rebuttal · Authors · 2023-08-29
>
> Thank you for your comprehensive review and for expressing interest in our work. We’ve considered your comments carefully and provide the following responses:
>
> **1A. The authors took the categories from Golovneva et al. (2022) and made a new classification for defining feedback types. I wonder if the different types of feedback have different effects on performance improvement. And, what additional kinds of feedback can be added to improve the performance of different tasks using LLMs?**
>
> > Different types of feedback do result in different performance improvements. One of the main strengths of our framework is that it leverages specific feedback in several error categories rather than a single general feedback such as in Self-Refine (Madaan et al. 2023). This results in performance improvement compared to Self-Refine as shown in Table 2.
> Furthermore, different feedback modules have varying effects on the performance. We have conducted an ablation study in which we removed feedback modules one by one from the feedback framework and tested the performance on a subset of the GSM-IC dataset, with results shown in Table 3. It can be seen that for this dataset, redundancy has the largest impact on performance while variable naming and missing step have a much smaller impact.
> >
> > For future work, there are many possible feedback modules that can be added to improve LLM performance. Notably, in the future, we would like to add external tools such as a calculator or entailment checker that can provide more concrete feedback on generated answers. Furthermore, the benchmarks and feedback modules in this paper focus on logical reasoning and question-answering. For generative tasks, feedback modules that focus on conciseness, grammar, style, informativeness, etc. may help generate higher-quality texts.
>
> **1B. What other NLP tasks can be benefited from MAF? And why do you think so?**
>
> > We believe that MAF is particularly applicable to problems that may be solved by generating reasoning chains. This is because MAF is able to examine individual reasoning steps and determine if they fall into any of the error categories as well as suggest specific fixes for it.
> >
> > One area that we would like to explore using MAF in scenarios where an LLM generates new content from a prompt, such as dialogue generation or summarization. As mentioned above, these tasks could benefit from feedback and refinement specifically targeted towards categories such as conciseness, grammar, style, informativeness, etc. depending on what the user prioritizes in their response.
>
> **1C. Could you explain more about over-refinement problem and in what conditions the problem occurs and how your MAF can mitigate the problem?**
>
> > Over-refinement occurs when the model reaches a correct solution through iterative refinement but cannot successfully detect to stop correcting the solution. In this case, feedback with a spurious error is generated and results in an incorrectly refined solution.
> >
> > To clarify, MAF does not mitigate the problem and we notice that our framework is also susceptible to the Over-Refinement problem. In fact, the interplay of different feedback modules exacerbates the problem as we increase the number of refinement steps. For this reason, we report all MAF results under the standard setting after 2 iterations, as compared to after 4 iterations for Self-Refine. The number of iterations is treated as a hyperparameter.
> However, the Self-Refine framework also suffers from the problem on the DROP dataset, which results in a drop in performance after using Self-Refine.
> >
> > We wanted to highlight the problem of Over-Refinement to encourage future works to study the Over-Refinement problem in iterative frameworks more closely.
>
> Additionally, we would like to acknowledge that we have considered all items in the ACL Responsible NLP checklist. We hope this response adequately addresses your queries. Thank you once more for your constructive feedback.

---

### Meta-Review · Area_Chair_ziJj · 2023-09-18

**Recommendation:** 4

**Metareview:**

This paper focuses on the problems of integrating multiple feedback modules (including frozen LMs and external tools) for improving reasoning in LLMs.

The reviewers agree that the problem is interesting and the method is effective. The one major weakness is that the ablation studies suggest that some modules have small to no effect on the final results. Overall, this is above the threshold for acceptance.

---

### Decision · Program_Chairs · 2023-10-07

**Decision:**

Accept-Main

**Comment:**

This paper focuses on the problems of integrating multiple feedback modules (including frozen LMs and external tools) for improving reasoning in LLMs.

The reviewers agree that the problem is interesting and the method is effective. The one major weakness is that the ablation studies suggest that some modules have small to no effect on the final results. Overall, this is above the threshold for acceptance.